# VIDEO GENERATORS ARE ROBOT POLICIES

## ABSTRACT

Despite tremendous progress in dexterous manipulation, current visuomotor policies remain fundamentally limited by two challenges: they struggle to generalize under perceptual or behavioral distribution shifts, and their performance is constrained by the size of human demonstration data. In this paper, we use video generation as a proxy for robot policy learning to address both limitations simultaneously. We propose Video Policy, a modular framework that combines video and action generation that can be trained end-to-end. Our results demonstrate that learning to generate videos of robot behavior allows for the extraction of policies with minimal demonstration data, significantly improving robustness and sample efficiency. Our method shows strong generalization to unseen objects, backgrounds, and tasks, both in simulation and the real world. We further highlight that task success is closely tied to the generated video, with action-free video data providing critical benefits for generalizing to novel tasks. By leveraging large-scale video generative models, we achieve superior performance compared to recent VLAs and video-action models, paving the way for more scalable and data-efficient robot policy learning.

## 1 INTRODUCTION

The fundamental challenge in visuomotor policy learning today is to create a robot system that robustly generalizes to new situations. Methods such as behavior cloning (Bain & Sammut, 1995; Chi et al., 2023) have achieved impressive performance in many tasks ranging from pick-and-place operations to complex dexterous manipulation (Brohan et al., 2022; Team et al., 2024; Black et al., 2025). However, state-of-the-art methods are often unable to transfer learned behaviors to novel situations, failing to generalize across distribution shifts ranging from simple variations (e.g., color changes) to more complex challenges such as adapting to entirely new tasks (Ross et al., 2011; Xiao et al., 2022; Barreiros et al., 2025). While fields like computer vision and natural language processing have tackled these issues by collecting increasingly larger training sets to capture all variations (Deng et al., 2009; Wenzek et al., 2019; Radford et al., 2021), robot action and human demonstration data are expensive to collect, making real-world generalization challenging.

Generative video models (Blattmann et al., 2023a), such as OpenAI's Sora (Brooks et al., 2024), are a promising direction for addressing these generalization challenges. Trained on massive datasets capturing diverse dynamics, situations, and physical interactions, they potentially encode powerful priors about how objects move and how actions affect the world. Thus, there is a strong interest in extracting robot policies from these foundation models to create robust control systems that can generalize to diverse scenarios and tasks. However, prior work along this direction has faced a fundamental trade-off: methods either rely on hand-crafted decoding mechanisms (such as tracking) (Liang et al., 2024) that limit expressivity and fail to capture the model's full knowledge, or they use learned action decoders that introduce their own generalization gaps due to limited demonstration data (Du et al., 2023b). Very recently, Hu et al. (2025) have proposed a flexible approach for combining video generation and policy learning, but their work lacks a detailed analysis of the interplay between the two objectives.

This paper demonstrates that video generative models are robot policies that generalize behavior to both visual and task distribution shifts. Our proposed model, **Video Policy**, is illustrated in Figure 1. We systematically study video diffusion models and demonstrate that, as long as video generative models synthesize accurate videos of robot behavior, then learned decoders only need a surprisingly small amount of demonstration data to learn to map videos into actions that a robot can directly

Figure 1: **Video Generation as a Proxy for Robot Policy Learning.** Given an initial observation and a natural language task prompt, Video Policy generates a video of a robot executing the task (top) jointly with generating robot actions via a separate diffusion network (middle). This modular design enables learning from action-free video data and improves generalization to unseen scenarios, offering a scalable and sample-efficient alternative to traditional behavior cloning.

execute. Remarkably, we find that a small decoder can be trained to generalize to unseen tasks, and that training the video generative model independently from the action decoder leads to substantial performance gains. These findings suggest that the video generative model is serving as the policy, while the decoder primarily serves as an interface rather than learning the task policy itself.

Since the policy relies heavily on videos synthesized by a large-scale generative model, and this video model has been trained on a diverse set of videos, our approach generalizes to novel objects, scenes, and tasks with less training data than existing methods. In both simulation and real-world experiments, we demonstrate that leveraging strong video priors improves performance compared to recent large-scale VLAs (Black et al., 2025; Pertsch et al., 2025; Bjorck et al., 2025) as well as custom-designed video-action models (Li et al., 2025). Furthermore, through detailed ablation studies, we provide insights into crucial design choices, offering guidance for future research on exploiting generative video models for robot learning. We will release our code, fine-tuned models and implementation details to ensure reproducibility.

## 2    RELATED WORKS

**Behavior Cloning.** Behavior cloning (BC) is a widely adopted method in manipulation tasks, where policies are learned from demonstrations using supervised learning. Initial BC approaches focused on end-to-end models that mapped states directly to actions, yet these methods often encountered difficulties with multimodal behavior and tasks that require high precision (Pomerleau, 1988; Zhang et al., 2018; Florence et al., 2019). To overcome these challenges, later research investigated Energy-Based Models (EBMs), which determine actions by minimizing an energy function during sequence optimization (LeCun et al., 2006; Du & Mordatch, 2019; Florence et al., 2022; Huang et al., 2023). More recently, conditional generative models have emerged as a promising alternative, effectively capturing diverse demonstration behaviors and improving task success rates (Chi et al., 2023; Zhao et al., 2023; Lee et al., 2024). Unlike earlier models, our approach is based on video diffusion models with an extra action diffusion head to predict robot action jointly with pixels.

**Visual Pretraining for Policy Learning.** There has been extensive research on pre-training perception models within visuomotor policies to achieve more robust visual representations. One commonly used objective is video prediction, where the model learns to forecast future frames from current observations, thereby capturing the dynamics and causal relationships essential for physical interactions (Finn et al., 2016; Sermanet et al., 2018; Babaeizadeh et al., 2018; Lee et al., 2018; Suris et al., 2021). Other popular self-supervised techniques include contrastive learning (Sermanet et al., 2018; Laskin et al., 2020; Radford et al., 2021; Nair et al., 2022) and masked autoencoding (Seo et al., 2023; Radosavovic et al., 2023), which both contribute to developing strong visual features for robotics. Additionally, some studies have focused on learning a generalizable reward function

through visual pretraining to support reinforcement learning tasks (Chen et al., 2021; Ma et al., 2023; Escontrela et al., 2023).

**Video Models for Decision-Making.** Early contributions in video generation (Ranzato et al., 2014; Vondrick et al., 2016; Finn et al., 2016) set the stage for employing generative models to predict future frames in a sequence. With recent advances in text-to-video generative models (Ho et al., 2022; Blattmann et al., 2023a;b; Zhang et al., 2023; Brooks et al., 2024), there is renewed interest in harnessing internet-scale video data for robotics. One line of research leverages video generative models as a form of world simulation, predicting future video sequences conditioned on actions (Yang et al., 2023; Du et al., 2023b; Brooks et al., 2024). Meanwhile, another approach utilizes video-language models to aid in long-horizon planning (Du et al., 2023a; Ajay et al., 2023; Black et al., 2023). Several works also explore the joint pixel and action diffusion architecture (Hu et al., 2025; Cheang et al., 2024; Guo et al., 2025; Li et al., 2025; Zhu et al., 2025). Compared to these concurrent works, we focus on providing detailed analysis and systematic evaluations.

## 3 METHODS

### 3.1 OVERVIEW

Given an input scene $v_0$ and a task description $c$, Video Policy generates a sequence of actions $a_t \in \mathbb{R}^k$ to accomplish the task, where $k$ is the action dimension of the robot's end-effector. We use a video generator $f$ that synthesizes videos of robot roll-outs $\{\hat{v}_t\}$ as the backbone for the policy, and a learned model $g$ to predict the robot actions from the synthesized frames:

$$\{\hat{v}_t\} = f(v_0, c) \tag{1}$$

$$\{a_t\} = g(\psi_0, \ldots, \psi_i) \quad \text{where} \quad \psi_i = f_i(v_0, c) \tag{2}$$

such that $f_i$ is the $i$th hidden layer of the video generator. This architecture is attractive because it integrates passive pre-training from internet videos to provide priors for generalization, and active demonstrations to learn strong policies in the physical world. We achieve this by fine-tuning $f$ and $g$ to generate videos and actions of the task being executed. At inference time, the generated actions are directly executed on a robot to perform manipulation tasks.

### 3.2 ARCHITECTURE

To integrate video generation with policy learning, we design denoising diffusion architectures $f$ and $g$ which jointly predict future video frames $\{\hat{v}_t\}$ and action sequences $\{a_t\}$. To achieve this, $f$ is a video U-Net $\mu_\theta$ and $g$ is an action U-Net $\alpha_\theta$. See Figure 2 for an overview.

Building off of the Image-To-Video Stable Video Diffusion (SVD) (Blattmann et al., 2023a) architecture, we condition the Video U-Net $\mu_\theta$ via cross-attention on the CLIP (Radford et al., 2021) embedding $\phi(c)$ of the textual description of the task $c$, and in the second stream we channel-wise concatenate a VAE-encoded image $z_0 = \text{VAE}(v_0)$ with the encoded noisy frames of the video $z_1, \ldots z_t$ (we use the frozen VAE from SVD).

We extend the architecture to decode actions using an action U-Net, $\alpha_\theta$, which is conditioned on intermediate features from the video denoising network. At each denoising step $i$, five evenly spaced hidden embeddings (layers 9, 14, 17, 20, 23) are extracted from the decoder layers of the video U-Net. These spatiotemporal features are passed through a CNN adapter, which transforms the latent embeddings into a single vector $h_i$ at denoising step $i$. This vector serves as a global conditioning input to a 1D CNN U-Net $\alpha_\theta$, adapted from Diffusion Policy (Chi et al., 2023), which then generates the sequence of robot actions $\{a_t\}$ via

$$\{a_t\} = \alpha_\theta(a_i, i, h_i). \tag{3}$$

This occurs per denoising step $i$, tightly integrating the video and action predicting, and allowing actions and videos to be generated simultaneously.

### 3.3 LEARNING

Video Policy is trained in two stages, where the video model is first trained for video prediction, and then the action model is trained for behavior cloning, with a dataset $D = d_1...d_n$ of expert demon-

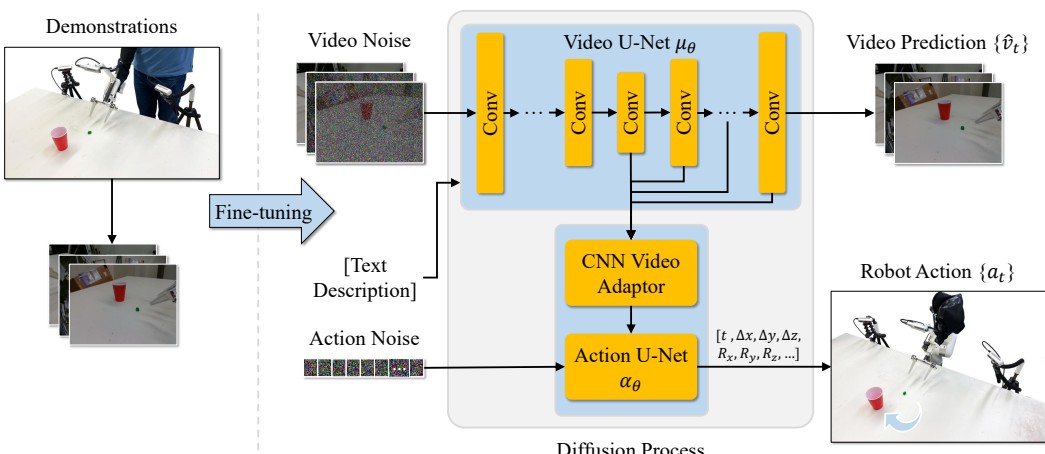

Figure 2: Video Policy takes an image of the initial environment state together with the noise vectors corresponding to the future frames and actions as input. It then jointly diffuses the frames and actions, using the representation of the frames as conditioning for the action denoiser. This modular design allows for training the two networks separately, opening way for action-free learning of the task dynamics via video generation.

strations. Each demonstration $d_i$ includes a video observation of the scene $\{v_t\}$, text description of the task $c$, and the corresponding robot actions $\{a_t\}$.

Our video model, $\mu_\theta$ is trained to minimize:

$$L_{video} = E_{z_0, \epsilon, i}[||\epsilon - \mu_\theta(z_i, i, \phi(c), z_{i,0})||^2], \tag{4}$$

where $i$ refers to a denoising time step, $z_{i,0}$ refers to the noisy latent embedding of the first frame, and $z_0$ refers to the denoised latent video.

Our action head, $\alpha_\theta$ is trained to minimize:

$$L_{action} = E_{a_0, \epsilon, i}[||\epsilon - \alpha_\theta(a_i, i, h_i)||^2], \tag{5}$$

where $h_i$ refers to the feature vector from our CNN adaptor at noise level $i$.

Since we want the video network to drive the policy due to its extensive pre-training, and $\alpha_\theta$ to just decode the robot action, we stop the gradients from $L_{action}$ from propagating back to $\mu_\theta$. As we show in the experiments, this leads to significant improvements in performance.

Once Video Policy is trained, we deploy it by providing an initial visual observation of the scene $v_0$ and task description $c$. The model then generates a sequence of predicted video frames $\{\hat{v}_t\}$ jointly with a corresponding sequence of robot actions $\{a_t\}$. The predicted actions $\{a_t\}$ are subsequently used to directly control the robot's end-effector to execute the manipulation task.

## 4 Experiments

### 4.1 Implementation Details

We use three cameras (one mounted on the gripper and two on either side of the scene) as video observation input. The three views are concatenated along the temporal dimension, such that $v_t \in R^{t,c,h,w}$, and the model is trained to predict 8 frames per camera view (24 frames in total). For Libero10 benchmark we adopt a slightly different setup to train using the agent and gripper camera views to predict 12 frames per camera view. To align with the pretrained SVD model's expected input, we pad a single camera frame to the start of the sequence, resulting in a total of 25 frames for all models. During training, we set a constant learning rate for video prediction to $1e-5$ and action prediction to $5e-5$.

Table 1: Comparison to the state of the art on the validation set of RoboCasa using success rate over 50 roll-outs per task. Video Policy using 50 human demonstrations is able to achieve state-of-the-art results on average and across most individual tasks. Training Video Policy on a larger set of 300 MimicGen demonstrations further improves its performance. Baselines for 3DA, DP3 and FPV are adopted from Donat et al. (2025), and results for GR00T and DP-VLA are reported from their respective papers.

| Category | Task | 3DA | DP3 | DP-ResNet | DP-CLIP | GR00T | FPV | DP-VLA | UVA | Video Policy 50 demos | Video Policy 300 demos |
|---|---|---|---|---|---|---|---|---|---|---|---|
| Pick and Place | PnPCabToCounter | 0.00 | 0.04 | 0.06 | 0.00 | 0.20 | 0.10 | 0.10 | 0.26 | 0.36 | **0.48** |
| | PnPCounterToCab | 0.00 | 0.02 | 0.06 | 0.02 | 0.36 | 0.14 | 0.32 | 0.18 | 0.42 | **0.52** |
| | PnPCounterToMicrowave | 0.00 | 0.06 | 0.06 | 0.02 | 0.13 | 0.10 | **0.56** | 0.10 | 0.52 | 0.22 |
| | PnPCounterToSink | 0.00 | 0.00 | 0.14 | 0.08 | 0.10 | 0.08 | 0.30 | 0.16 | 0.44 | **0.48** |
| | PnPCounterToStove | 0.00 | 0.00 | 0.00 | 0.02 | 0.24 | 0.04 | 0.22 | 0.16 | **0.58** | 0.54 |
| | PnPMicrowaveToCounter | 0.00 | 0.06 | 0.06 | 0.06 | 0.16 | 0.12 | 0.18 | 0.18 | **0.44** | 0.28 |
| | PnPSinkToCounter | 0.00 | 0.00 | 0.10 | 0.22 | 0.33 | 0.30 | 0.56 | 0.38 | **0.64** | 0.56 |
| | PnPStoveToCounter | 0.00 | 0.00 | 0.02 | 0.06 | 0.29 | 0.26 | 0.62 | 0.24 | 0.64 | **0.70** |
| Doors | OpenSingleDoor | 0.00 | 0.24 | 0.42 | 0.32 | 0.59 | **0.74** | 0.42 | 0.54 | 0.68 | 0.70 |
| | OpenDoubleDoor | 0.00 | 0.20 | 0.70 | 0.82 | 0.15 | 0.92 | 0.80 | 0.90 | **0.96** | 0.94 |
| | CloseDoubleDoor | 0.00 | 0.56 | 0.78 | 0.84 | 0.75 | 0.78 | 0.84 | 0.76 | **0.98** | 0.90 |
| | CloseSingleDoor | 0.14 | 0.62 | 0.78 | 0.48 | 0.83 | 0.84 | **1.00** | 0.88 | 1.00 | 0.90 |
| Drawers | OpenDrawer | 0.00 | 0.36 | 0.64 | 0.60 | **0.79** | 0.72 | 0.66 | 0.28 | 0.46 | 0.54 |
| | CloseDrawer | 0.00 | 0.48 | 0.82 | 0.96 | 0.99 | 0.94 | **1.00** | 0.72 | 0.96 | **1.00** |
| Twisting Knobs | TurnOnStove | 0.10 | 0.24 | 0.38 | 0.28 | 0.56 | **0.66** | 0.64 | 0.50 | 0.30 | 0.50 |
| | TurnOffStove | 0.02 | 0.06 | 0.16 | 0.08 | **0.27** | 0.20 | 0.16 | 0.14 | 0.06 | 0.04 |
| Turning Levers | TurnOnSinkFaucet | 0.06 | 0.32 | 0.66 | 0.66 | 0.63 | 0.70 | 0.56 | 0.62 | **0.84** | 0.76 |
| | TurnOffSinkFaucet | 0.28 | 0.42 | 0.68 | 0.70 | 0.73 | 0.78 | 0.72 | 0.64 | 0.78 | **0.92** |
| | TurnSinkSpout | 0.26 | 0.54 | 0.62 | 0.26 | 0.53 | 0.52 | **0.90** | 0.64 | 0.40 | 0.58 |
| Pressing Buttons | CoffeePressButton | 0.08 | 0.16 | 0.76 | 0.68 | 0.85 | 0.90 | 0.86 | 0.84 | 0.92 | **0.96** |
| | TurnOnMicrowave | 0.06 | 0.38 | 0.68 | 0.88 | 0.78 | 0.68 | 0.84 | 0.94 | 0.92 | **0.96** |
| | TurnOffMicrowave | 0.32 | 0.54 | 0.62 | **1.00** | 0.71 | 0.96 | 0.86 | 0.96 | 0.90 | **1.00** |
| Insertion | CoffeeServeMug | 0.00 | 0.18 | 0.44 | 0.60 | 0.73 | 0.48 | 0.64 | **0.78** | 0.76 | 0.70 |
| | CoffeeSetupMug | 0.00 | 0.04 | 0.10 | 0.12 | 0.23 | 0.16 | 0.30 | 0.20 | 0.22 | **0.58** |
| **Avg. Task Success Rate** | | 0.06 | 0.23 | 0.41 | 0.43 | 0.50 | 0.51 | 0.57 | 0.50 | 0.63 | **0.66** |

Table 2: Average success rates for tasks in the Libero10 benchmark. Video Policy achieves the highest overall performance compared to baselines adopted from Li et al. (2025). Per-task results are reported in the supplementary.

| Model | DP-C | DP-T | OpenVLA | UniPi | $\pi_0$ | $\pi_0$-FAST | UVA | Video Policy |
|---|---|---|---|---|---|---|---|---|
| Avg. Task Success Rate | 0.53 | 0.58 | 0.54 | 0.00 | 0.85 | 0.60 | 0.90 | **0.94** |

## 4.2 Simulation Experimental Setup and Baselines

We perform quantitative evaluation of Video Policy using the RoboCasa (Nasiriany et al., 2024) and Libero10 (Liu et al., 2023) simulation benchmarks, which span a total of 34 manipulation tasks. For each task, both benchmarks provide 50 human demonstrations. In the RoboCasa benchmark, demonstrations are replayed at a resolution of 256×256 pixels. For Libero10, the demonstrations are resized to the same resolution.

The action space of the simulation benchmarks is defined as $a_i \in \mathbb{R}^7$, which includes the 6-DoF pose of the gripper and a scalar value representing the gripper's open or closed state. For RoboCasa evaluation, we follow the evaluation protocol outlined by Nasiriany et al. (2024); specifically, each task is evaluated over a total of 50 rollouts executed in five different RoboCasa scenes. For Libero10 evaluation, we follow the evaluation protocol in Li et al. (2025).

As RoboCasa baselines, we train Unified Video Action (UVA) (Li et al., 2025) and (ImageNet pre-trained) ResNet- and CLIP-based variants of Diffusion Policy on the same dataset, using identical inputs and evaluation environments as Video Policy. For additional RoboCasa and Libero10 baselines, we compare to the results reported from several prior works (Donat et al., 2025; Bjorck et al., 2025; Han et al., 2024; Li et al., 2025).

### 4.3 QUANTITATIVE RESULTS

We begin by comparing Video Policy to the state-of-the-art on the validation set of the synthetic RoboCasa benchmark and report the results in Table 1. Firstly, we observe that Video Policy outperforms the baselines by a large margin on average, as well as on most individual tasks. The improvements are especially noticeable on Pick and Place tasks which feature a significant distribution shift between training and test environments, highlighting the robustness of our approach. Methods capitalizing on explicit 3D representation of the environment (Ke et al., 2024; Ze et al., 2024), including 3DA (Ke et al., 2024), DP3 (Ze et al., 2024) and recent FPV (Donat et al., 2025), struggle to achieve competitive performance on this benchmark requiring rich semantic understanding. Diffusion Policy (Chi et al., 2023) demonstrates robust performance, but fails to scale when equipped with a strong CLIP (Radford et al., 2021) visual-language representation (columns 5, 6 in Table 1).

Most notably, Video Policy demonstrates competitive performance to baselines that capitalize on large-scale visual-language (DP-VLA from Han et al. (2024)) and video pre-training (GR00T from Bjorck et al. (2025)). In addition, both of these methods use more RoboCasa demonstrations for behavior cloning, with GR00T using 300 and DP-VLA 3000 automatically generated demonstrations per task via MimicGen (Mandlekar et al., 2023). In contrast, Video Policy requires only 50 demonstrations per task to achieve state-of-the-art results by effectively aligning the video and action generation objectives. Training Video Policy on more demonstrations further improves its performance (last column in the table).

Finally, the concurrent UVA (Li et al., 2025) approach, which also proposes a joint model for video and action prediction, fails to generalize to the challenging RoboCasa benchmark due to its over-reliance to the single-camera setup. Specifically, UVA decodes a joint latent with a Transformer that couples action chunks with future-frame tokens, restricting predictions to one camera stream at a time. In contrast, our straightforward architecture can easily support any configuration of the environment. We also compare Video Policy to UVA and other strong baselines on the Libero-10 benchmark in Table 2, where we achieve significantly higher average task success rates, consistent with our results on RoboCasa benchmark. We analyze the key factors behind the effectiveness of our approach in detail next.

### 4.4 ANALYSIS

In this section, we set out to explore the interplay between the video and action generation. To this end, we first investigate the effect of isolating the two training objectives in Equations 4 and 5 compared to joint training. In joint training we fine-tune the model with both training objectives, whereas in 2-stage training we first fine-tune SVD for video generation on the training set of RoboCasa, then freeze the video diffusion U-Net weights, and learn the action denoising head on top of this frozen representation. Remarkably, as shown in row 2 in Table 3, the 2-stage variant has a significantly

Table 3: Ablation study on the RoboCasa validation set, analyzing the interplay between video and action generation. Results show that learning to generate policy-execution videos is both necessary and sufficient for learning robust manipulation representations.

| Model Variant | Avg. Task Success Rate |
|---|---|
| Joint | 0.57 |
| 2-Stage | **0.63** |
| No Video Tuning | 0.09 |

higher performance in terms of average success rate compared to the end-to-end trained model (denoted as 'Joint' in the table, per-task results are reported in the supplementary). This result suggests that learning to generate videos in the raw pixel space is a strictly more general objective than action generation.

Next, we study the properties of the video generation objective in greater detail. Firstly, we explore the importance of fine-tuning SVD on RoboCasa in the last row of Table 3. Our results indicate that learning to generate videos of the robot executing a policy is a crucial component of our training pipeline, whereas vanilla SVD pretraining is insufficient for the task. This finding further supports our earlier observation that video generation of policy execution serves as a strong proxy for policy learning. In Figure 3, we show that the prediction horizon of the video generation model is a key factor in the effectiveness of this proxy objective. All model variants predict actions 1.6 seconds

into the future, while the video prediction horizon is varied ranging from 0 (i.e., reconstructing the conditioning frame) to longer rollouts. We plot the average task success rates on RoboCasa — separately for tasks with and without distribution shift between training and evaluation environments — as a function of the video prediction horizon used during training (distribution shift details and per-task results are provided in the supplementary). We roll out action prediction for 0.8 seconds across all variants. While longer-term video prediction universally improves performance, the effect is more significant for tasks that require stronger generalization. These results highlight that learning accurate environment dynamics is critical for achieving generalization in policy learning.

Action-free videos can provide a nearly limitless data source for learning the environment dynamics. To demonstrate their utility, we start from SVD fine-tuned for video generation on the entire training set of RoboCasa (Stage 1 from the 2-Stage variant in Table 3), but learning the action denoising head in Equation 3 on only half of the tasks sampled at random. We evaluate the resulting model on the full set of 24 tasks and compare to the DP-ResNet baseline trained on the same 12 tasks in Figure 4. The results demonstrate that Video Policy can achieve strong generalization to the unseen tasks during policy training (right half of the figure) by capitalizing on action-free video generation pre-training. In contrast, the DP-ResNet baseline, which is not able to utilize action-free data, only exhibits a minimal degree of generalization to unseen tasks.

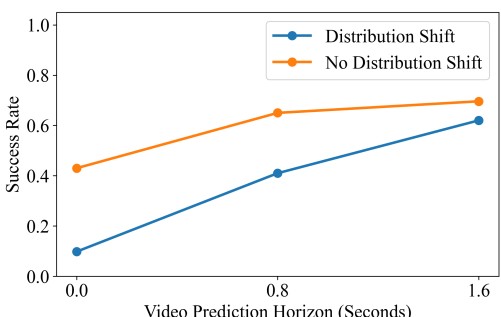

Figure 3: Success rate of Video Policy as a function of the video prediction horizon. Learning the dynamics of the environment is critical for achieving generalization in policy learning, as evident by the larger effect of prediction horizon on the task with distribution shift.

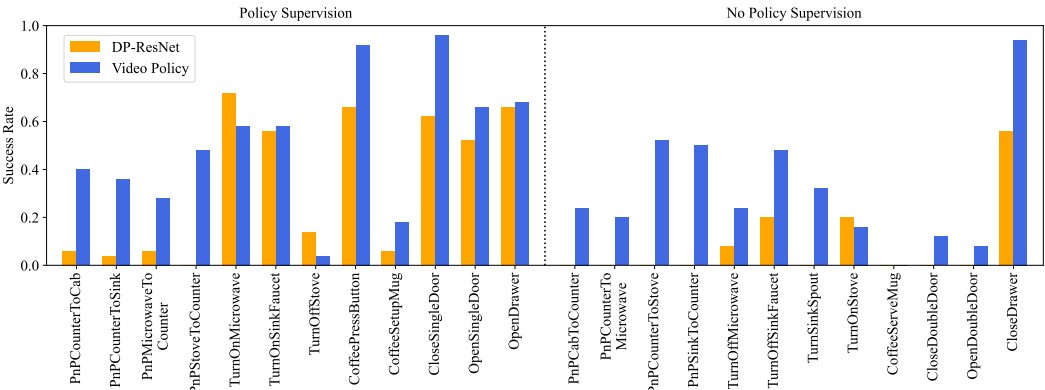

Figure 4: Generalization to tasks with no policy supervision by capitalizing on action-free video data. Both our behavior cloning head and the baseline DP are trained on 12 tasks on the left, but our video generation model also has access to action-free videos for all 24 tasks. We demonstrate that this additional data enables Video Policy to generalize to the unseen tasks on the right solely through video generation training, without any action decoder supervision. In contrast, the DP ResNet baseline is unable to benefit from the action-free video data and shows limited generalization to the novel tasks.

## 4.5 REAL-WORLD RESULTS

In this section, we further validate the generalization advantages of policy learning via video generation in the real world. Our evaluation encompasses five tasks: Open Drawer, Pick and Place, M&Ms to Cup, Upright Object and Stack Cups. The details of the evaluation are discussed below.

**Robot Setup.** For the five tasks, human demonstrations were collected using a handheld gripper equipped with the same end effector as the robot. The setup closely mimics the RoboCasa camera

configuration. During each demonstration, a human operator performs the task with the gripper while RGB video is recorded, along with action states of the gripper — including its 6D pose, parallel jaw opening width, and the force exerted by the jaws. The gripper pose is tracked using an Intel RealSense T265 camera, and the gripping force is measured with a uniaxial load cell. The data collection and the robot execution setup are illustrated in Figure 2. For each task, 200 demonstrations are collected for training.

**Experiment Setup.** We trained Video Policy following the simulation pipeline from the SVD checkpoint and evaluated our model in 5 manipulation tasks, using a suite of experiments designed to test its generalization with variations in object location, interacting with unseen objects, and changes in background appearance. Details of the training and testing object sets for each task are provided in the supplementary material.

The tasks are as follows: Open Drawer involves the robot grasping a drawer handle and opening it to over 50% of its full extension, with two drawers at different heights. Pick and Place has the robot picking up an object from a cluttered table and placing it into a container, with 11 objects and 10 containers used. In M&Ms to Cup, the robot picks up an M&M and places it in a cup, with distractors present; the training set includes 6 M&M colors and 5 cups. Upright Object involves the robot repositioning an object on the table to stand upright, with 6 manipulable objects and distractors. Stack Cups has the robot stacking one cup into another, using 6 different cups on a cluttered table.

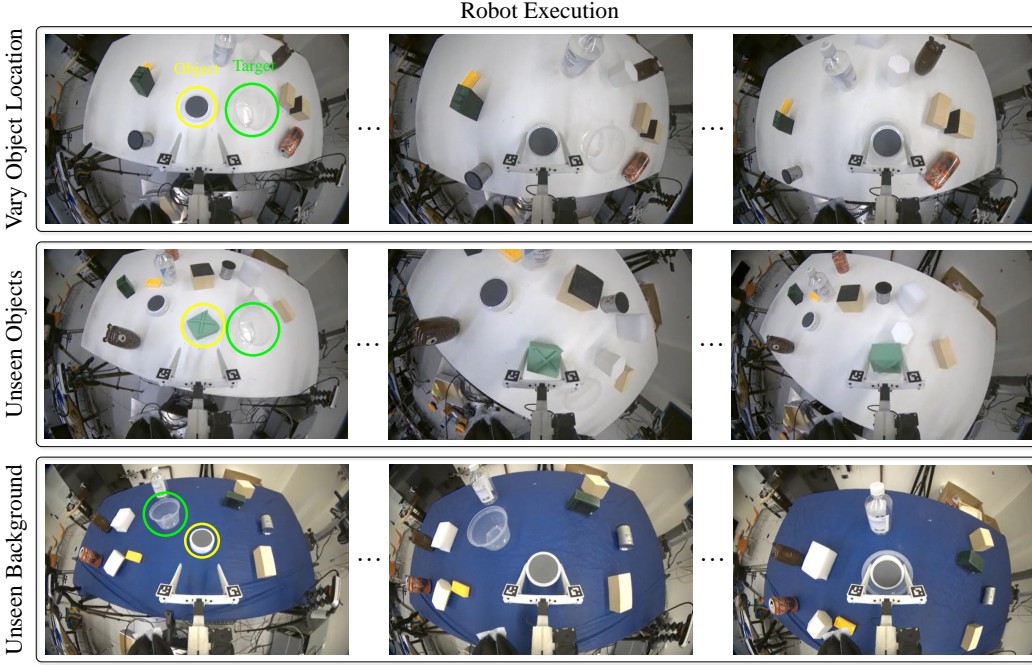

Figure 5: Qualitative results for Pick and Place generalization experiments in the real world. Video Policy demonstrates strong robustness to object locations, appearance and background colour.

## 4.6 GENERALIZATION ANALYSIS

We report success rates over 10 roll-outs in Table 4, with qualitative examples in Figure 5 and the supplementary material.

**Generalization to Object Locations.** In this experiment, we use the same objects as in the training demonstrations but exact positions can vary between training and testing, serving as a baseline with minimal distribution shift for comparison with other scenarios. We observe that the model performs best on the Open Drawer, Pick and Place, and M&Ms to Cup. A successful example on the Pick and Place task is shown in the first row of Figure 5, where the model accurately places an object into a difficult-to-see transparent cup. Failures in Upright Object and Stack Cups are due to unrealistic

video predictions — e.g., failing to generate upright placements or gripper-induced toppling — likely due to limited real-world physics priors in the pretrained SVD model.

**Generalization to Unseen Objects.** When tested on novel objects with different shapes or colors, the model maintains high performance across tasks. Figure 5 (second row) shows a successful example of the Pick and Place task with an irregularly shaped object not seen during training. Video Policy successfully adjusts the parallel-jaw gripper to find a suitable grasp and places the object into the target cup. Notably, the Upright Object task shows significant improvement, likely due to the test objects being opaque, in contrast to the transparent cups used during training. The results suggest that policy learning through video generation can improve robustness to novel objects.

**Generalization to Unseen Background.** Training on a white surface, we now test with black, red, or blue backgrounds. We find that Open Drawer, Pick and Place, Upright Object, and Stack Cups all maintain strong performance under these conditions. As shown in the last row in Figure 5, the model successfully completes the Pick and Place task even when the table is covered with an unseen blue background. Once again, Upright Object shows improved performance likely because the colored backgrounds enhance the visibility of the transparent cups. In contrast, performance on the M&Ms to Cup task decreases. The robot gripper often fails to accurately localize the M&Ms, suggesting that background color changes can adversely affect robot actions that require high precision.

Table 4: Generalization performance in real world. Success rates computed over 10 roll-outs per task across three experiments, evaluating different generalization dimensions: object location, object appearance, and background appearance. Video Policy shows strong robustness across all three dimensions in the real world.

| Tasks | Vary Object Location | Unseen Objects | Unseen Background |
|---|---|---|---|
| Open Drawer | 0.8 | 1.0 | 0.9 |
| Pick and Place | 1.0 | 0.9 | 0.8 |
| M&Ms to Cup | 0.8 | 0.9 | 0.2 |
| Upright Object | 0.3 | 0.7 | 0.8 |
| Stack Cups | 0.3 | 0.2 | 0.2 |

## 5 CONCLUSION

In this work, we investigate the interplay between video and action diffusion objectives for policy learning. Our results show that generating pixels can serve as an effective proxy for learning policy, substantially improving the robustness and generalization of behavior cloning. Moreover, we observe that separating the video generation objective from action generation substantially improves performance, and a lightweight decoder can generalize action decoding to unseen tasks. These findings suggest that the video generative model itself functions as the policy, while the decoder primarily serves as an action decoder rather than the policy. A key insight from our study is that casting policy learning as video generation unlocks the ability to leverage action-free data — broadening the scope of usable training signals. As generative models continue to scale with massive in-the-wild video datasets, this paradigm offers a promising path toward more scalable and generalizable policy learning for real-world manipulation.

## 6 LIMITATIONS

Our study has several limitations. First, it is restricted in the scale of simulation benchmarks and a single real-world embodiment. Additionally, we explore only one instantiation of video generation models — Stable Video Diffusion. While our analysis is more extensive than prior works, broader validation across tasks, environments, and model families would further strengthen the findings. Second, the computational cost of video diffusion models remains a major practical bottleneck, particularly for real-world deployment. However, recent advances in accelerating diffusion inference (Song et al., 2023; Esser et al., 2024; Zhou et al., 2024) offer a promising path toward real-time performance, which could unlock broader applicability in robotics.

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

# A APPENDIX

## A.1 VIDEO MODEL IMPLEMENTATION

We adapted the pretrained SVD model, which generates 25-frame video sequences. In our Robo-Casa Experiments, frame 1 is a padded frame, frames 2–9 correspond to the gripper view, frames 10–17 to the left camera view, and frames 18–25 to the right camera view. To enable the model to generate such multi-view videos, we modified the per-frame image embedding based on the viewing camera of each output frame. By default, the generated video contains 8 frames for each view and represents the robot's actions in default RoboCasa action space over 32-steps prediction horizon (the videos are subsampled using a stride of 4). This setup is consistent across both simulation and real-world experiments. Training hyperparameters are detailed in Table 5. We fine-tuned the model on eight A100 GPUs over approximately two weeks and observed that training any further did not yield performance improvements. For the real-world model, we initially trained at a resolution of $256 \times 192$ to speed up training, followed by training at a higher resolution of $448 \times 320$ to improve performance. During inference for all experiments, we use 30 denoising steps with a constant classifier-free guidance scale of 2.0. On an A100 GPU, a 25 frame video takes around 9 seconds to generate with a resolution of $256 \times 256$ and 30 diffusion steps.

All RoboCasa experiments follow the standard RoboCasa evaluation protocol, except for our study on video prediction horizons. Per-task results under the standard protocol are shown in Table 6. For the video prediction horizon analysis in Figure 2, we adopt a different protocol to isolate the effect of distribution shift. Specifically, we aimed to investigate how per-task performance is affected by distribution shift as a function of the prediction horizon. To isolate this effect, we do not use the standard RoboCasa protocol where all evaluation environments are out-of-distribution and instead evaluate on sampled environments from the MimicGen dataset. These environments share the same layout and style as the training set but differ in object positions and object categories through synthetic generation. In particular, the pick-and-place tasks involve the greatest variation in object position and type. Figure 6 compares training and evaluation environments for two example tasks: Open Single Door and PnP Counter to Cabinet. While the door type and position vary minimally in Open Single Door, the PnP Counter to Cabinet task involves substantial differences in object types, illustrating the degree of distribution shift. We are interested in comparing the performance of pick-and-place tasks, which exhibit significant distribution shift, to other tasks with minimal shift, across varying prediction horizons. The corresponding per-task results are presented in Table 8.

Table 5: Hyperparameters for video model training. Resolution: image and video resolution, Lr: learning rate, Batch: batch size, Steps: training steps for the evaluation checkpoint, Precision: lightning trainer precision.

| Models | Resolution | Lr | Batch | Steps | Precision |
|---|---|---|---|---|---|
| Joint Training | $256 \times 256$ | 1e-5 | 32 | 368866 | 16-mixed |
| 2-Stage Training | $256 \times 256$ | 1e-5 | 32 | $368866 \times 2$ | 16-mixed |
| No Video Tuning | $256 \times 256$ | 1e-5 | 32 | 368866 | 16-mixed |
| Joint Training (16 Steps) | $256 \times 256$ | 1e-5 | 32 | 368866 | 16-mixed |
| Joint Training (0 Steps) | $256 \times 256$ | 1e-5 | 32 | 368866 | 16-mixed |
| 2-Stage Training (Half Tasks) | $256 \times 256$ | 1e-5 | 32 | $368866 \times 2$ | 16-mixed |
| 2-Stage Training (Libero10) | $256 \times 256$ | 1e-5 | 32 | 170000+140000 | 16-mixed |
| Joint Training (Real World) | $256 \times 192 \rightarrow 448 \times 320$ | 1e-5 | 32 | 331500+92960 | 16-mixed |

Table 6: Per-task success rates out of 50 trials across different experimental variants following the RoboCasa evaluation protocols. Joint: end-to-end trained model jointly predicting video and actions. 2-Stages: video and action predictions are trained separately. No Video: frozen video U-Net initialized with SVD pretraining. Half Tasks: video and action models trained separately, with the action denoising head trained on only half of the tasks. DP Half Tasks: ResNet-based CNN Diffusion Policy trained on only half of the tasks.

| Category | Task | Joint | 2-Stages | No Video | Half Tasks | DP Half Tasks |
|---|---|---|---|---|---|---|
| **Pick and Place** | PnPCabToCounter | 0.12 | 0.36 | 0.00 | 0.24 | 0.00 |
| | PnPCounterToCab | 0.38 | 0.42 | 0.00 | 0.40 | 0.06 |
| | PnPCounterToMicrowave | 0.42 | 0.52 | 0.00 | 0.20 | 0.00 |
| | PnPCounterToSink | 0.36 | 0.44 | 0.00 | 0.36 | 0.04 |
| | PnPCounterToStove | 0.54 | 0.58 | 0.00 | 0.52 | 0.00 |
| | PnPMicrowaveToCounter | 0.34 | 0.44 | 0.00 | 0.28 | 0.06 |
| | PnPSinkToCounter | 0.62 | 0.64 | 0.00 | 0.50 | 0.00 |
| | PnPStoveToCounter | 0.60 | 0.64 | 0.00 | 0.48 | 0.00 |
| **Doors** | OpenSingleDoor | 0.58 | 0.68 | 0.06 | 0.66 | 0.52 |
| | OpenDoubleDoor | 0.96 | 0.96 | 0.00 | 0.08 | 0.00 |
| | CloseDoubleDoor | 0.76 | 0.98 | 0.14 | 0.12 | 0.00 |
| | CloseSingleDoor | 0.96 | 1.00 | 0.28 | 0.96 | 0.62 |
| **Drawers** | OpenDrawer | 0.60 | 0.46 | 0.00 | 0.68 | 0.66 |
| | CloseDrawer | 0.96 | 0.96 | 0.10 | 0.94 | 0.56 |
| **Twisting Knobs** | TurnOnStove | 0.38 | 0.30 | 0.10 | 0.16 | 0.20 |
| | TurnOffStove | 0.12 | 0.06 | 0.06 | 0.04 | 0.14 |
| **Turning Levers** | TurnOnSinkFaucet | 0.40 | 0.84 | 0.34 | 0.58 | 0.56 |
| | TurnOffSinkFaucet | 0.66 | 0.78 | 0.20 | 0.48 | 0.20 |
| | TurnSinkSpout | 0.32 | 0.40 | 0.20 | 0.32 | 0.00 |
| **Pressing Buttons** | CoffeePressButton | 0.92 | 0.92 | 0.34 | 0.92 | 0.66 |
| | TurnOnMicrowave | 0.86 | 0.92 | 0.10 | 0.58 | 0.72 |
| | TurnOffMicrowave | 1.00 | 0.90 | 0.28 | 0.24 | 0.08 |
| **Insertion** | CoffeeServeMug | 0.68 | 0.76 | 0.06 | 0.00 | 0.00 |
| | CoffeeSetupMug | 0.12 | 0.22 | 0.00 | 0.18 | 0.06 |
| **Avg. Task Success Rate** | | 0.57 | 0.63 | 0.09 | 0.41 | 0.21 |

Table 7: Per-task success rates on the RoboCasa benchmark, comparing GR00T trained with varying numbers of demonstrations to Video Policy. The per-task GR00T results are reported from Bjorck et al. (2025).

| Category | Task | GR00T (Bjorck et al., 2025) | | | Video Policy |
| | | 30 demos | 100 demos | 300 demos | 50 demos |
|---|---|---|---|---|---|
| **Pick and Place** | PnPCabToCounter | 0.01 | 0.04 | 0.20 | 0.36 |
| | PnPCounterToCab | 0.02 | 0.07 | 0.36 | 0.42 |
| | PnPCounterToMicrowave | 0.00 | 0.00 | 0.13 | 0.52 |
| | PnPCounterToSink | 0.00 | 0.01 | 0.10 | 0.44 |
| | PnPCounterToStove | 0.00 | 0.00 | 0.24 | 0.58 |
| | PnPMicrowaveToCounter | 0.00 | 0.00 | 0.16 | 0.44 |
| | PnPSinkToCounter | 0.00 | 0.06 | 0.33 | 0.64 |
| | PnPStoveToCounter | 0.00 | 0.00 | 0.29 | 0.64 |
| **Doors** | OpenSingleDoor | 0.20 | 0.55 | 0.59 | 0.68 |
| | OpenDoubleDoor | 0.00 | 0.13 | 0.15 | 0.96 |
| | CloseDoubleDoor | 0.00 | 0.43 | 0.75 | 0.98 |
| | CloseSingleDoor | 0.49 | 0.68 | 0.83 | 1.00 |
| **Drawers** | OpenDrawer | 0.09 | 0.42 | 0.79 | 0.46 |
| | CloseDrawer | 0.77 | 0.96 | 0.99 | 0.96 |
| **Twisting Knobs** | TurnOnStove | 0.15 | 0.26 | 0.56 | 0.30 |
| | TurnOffStove | 0.05 | 0.16 | 0.27 | 0.06 |
| **Turning Levers** | TurnOnSinkFaucet | 0.33 | 0.60 | 0.63 | 0.84 |
| | TurnOffSinkFaucet | 0.49 | 0.68 | 0.73 | 0.78 |
| | TurnSinkSpout | 0.24 | 0.42 | 0.53 | 0.40 |
| **Pressing Buttons** | CoffeePressButton | 0.28 | 0.57 | 0.85 | 0.92 |
| | TurnOnMicrowave | 0.56 | 0.74 | 0.78 | 0.92 |
| | TurnOffMicrowave | 0.47 | 0.58 | 0.71 | 0.90 |
| **Insertion** | CoffeeServeMug | 0.04 | 0.34 | 0.73 | 0.76 |
| | CoffeeSetupMug | 0.00 | 0.02 | 0.23 | 0.22 |
| **Avg. Task Success Rate** | | 0.17 | 0.32 | 0.50 | 0.63 |

Table 8: Video Policy per-task success rates out of 50 trials for different video prediction horizon variants evaluated in sampled environments from the MimicGen dataset. Joint 32-Steps: joint video-action model trained with a 32-step video prediction horizon. Joint 16-Steps: same model trained with a 16-step video prediction horizon. Joint 0-Steps: model where the video prediction output is identical to the input images.

| Category | Task | Joint 32-Steps | Joint 16-Steps | Joint 0-Steps |
|---|---|---|---|---|
| **With Distribution Shift** | PnPCabToCounter | 0.28 | 0.24 | 0.14 |
| | PnPCounterToCab | 0.70 | 0.50 | 0.22 |
| | PnPCounterToMicrowave | 0.56 | 0.28 | 0.06 |
| | PnPCounterToSink | 0.56 | 0.42 | 0.06 |
| | PnPCounterToStove | 0.82 | 0.46 | 0.02 |
| | PnPMicrowaveToCounter | 0.58 | 0.30 | 0.02 |
| | PnPSinkToCounter | 0.62 | 0.52 | 0.12 |
| | PnPStoveToCounter | 0.80 | 0.56 | 0.14 |
| **Without Distribution Shift** | OpenSingleDoor | 0.92 | 0.88 | 0.26 |
| | OpenDoubleDoor | 0.98 | 0.96 | 0.58 |
| | CloseDoubleDoor | 0.80 | 0.74 | 0.40 |
| | CloseSingleDoor | 1.00 | 0.98 | 0.84 |
| | OpenDrawer | 0.48 | 0.34 | 0.18 |
| | CloseDrawer | 1.00 | 0.96 | 0.92 |
| | TurnOnStove | 0.46 | 0.18 | 0.22 |
| | TurnOffStove | 0.20 | 0.20 | 0.22 |
| | TurnOnSinkFaucet | 0.36 | 0.40 | 0.16 |
| | TurnOffSinkFaucet | 0.76 | 0.72 | 0.74 |
| | TurnSinkSpout | 0.60 | 0.70 | 0.72 |
| | CoffeePressButton | 0.96 | 0.92 | 0.22 |
| | TurnOnMicrowave | 0.80 | 0.48 | 0.38 |
| | TurnOffMicrowave | 0.80 | 0.42 | 0.22 |
| | CoffeeServeMug | 0.74 | 0.66 | 0.36 |
| | CoffeeSetupMug | 0.30 | 0.26 | 0.10 |
| **Avg. Task Success Rate** | | 0.67 | 0.55 | 0.30 |

Table 9: Video Policy per-task success rates out of 50 trials for the 10 tasks in Libero10 benchmark, following the evaluation protocol in UVA (Li et al., 2025).

| Libero10 Tasks | Success Rate |
|---|---|
| LIVING ROOM SCENE2 put both the alphabet soup and the tomato sauce in the basket | 0.96 |
| LIVING ROOM SCENE2 put both the cream cheese box and the butter in the basket | 1.00 |
| KITCHEN SCENE3 turn on the stove and put the moka pot on it | 1.00 |
| KITCHEN SCENE4 put the black bowl in the bottom drawer of the cabinet and close it | 0.98 |
| LIVING ROOM SCENE5 put the white mug on the left plate and put the yellow and white mug on the right plate | 0.94 |
| STUDY SCENE1 pick up the book and place it in the back compartment of the caddy | 0.96 |
| LIVING ROOM SCENE6 put the white mug on the plate and put the chocolate pudding to the right of the plate | 0.88 |
| LIVING ROOM SCENE1 put both the alphabet soup and the cream cheese box in the basket | 1.00 |
| KITCHEN SCENE8 put both moka pots on the stove | 0.80 |
| KITCHEN SCENE6 put the yellow and white mug in the microwave and close it | 0.86 |
| **Average** | 0.94 |

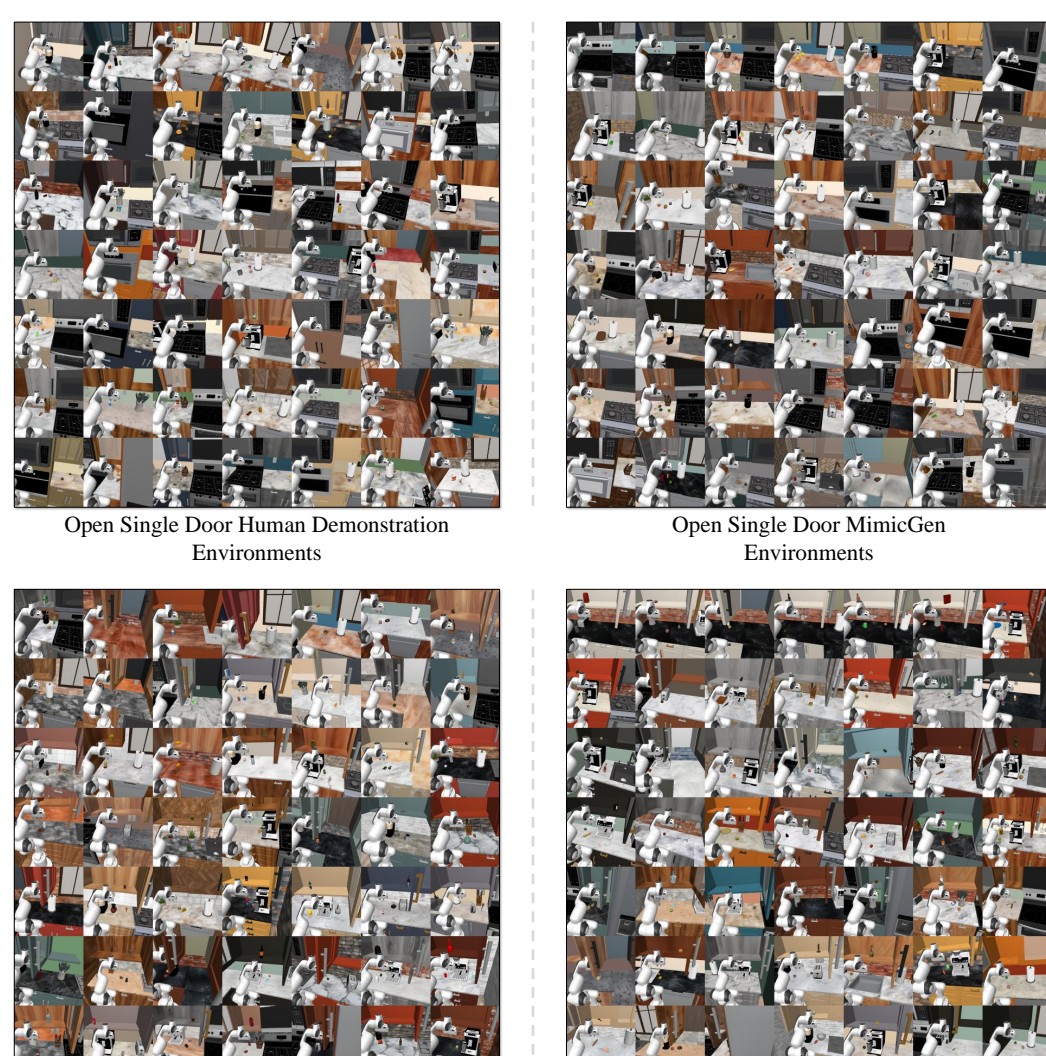

Open Single Door Human Demonstration
Environments

Open Single Door MimicGen
Environments

PnP Counter to Cabinet Human Demonstration
Environments

PnP Counter to Cabinet MimicGen
Environments

Figure 6: Example training and evaluation environments for the Open Single Door and PnP Counter to Cabinet tasks. Evaluation environments are sampled from the MimicGen dataset, which introduces variation in object types and positions for pick-and-place tasks such as PnP Counter to Cabinet, but have minimal variation for tasks like Open Single Door.

## A.2 UNIFIED VIDEO ACTION MODEL BASELINE

As a baseline, we train the Unified Video Action Model, initialized using a pretrained VAE and the MAR image generation model (Li et al., 2024). We modify the model to accept three images as conditioning frames and to generate all three camera views of the video at a resolution of 256×256, by concatenating frames along the temporal axis. The model is trained on the same dataset and evaluated under identical conditions as Video Policy.

## A.3 DIFFUSION POLICY BASELINE

As baselines, we train ResNet- and CLIP-based variants of the CNN Diffusion Policy using the implementation from UMI (Chi et al., 2024), both initialized with pretrained weights (ImageNet pre-training in case of ResNet). We use ResNet18 and CLIP-Base variants. The models are trained on the same dataset and evaluated under the same conditions as Video Policy. To match the text conditioning in the video model, we condition both variants on the task name using the same CLIP text encoder. The resulting text embedding is concatenated with the image observation embedding as global context. Each variant uses three separate encoders for the three camera views. For the ResNet-based model, input images are resized to 256×256, while the CLIP-based model uses 224×224 inputs. Both variants are trained with a batch size of 768 to predict 32 future steps and roll out 16 steps in simulation, matching the video model setup.

## A.4 REAL-WORLD EXPERIMENT SETUP

The demonstrations are recorded using RGB cameras mounted on the left, right, and on the gripper itself. The side-view RGB cameras are Intel RealSense D435, while the gripper-mounted camera is a Basler fisheye camera. The gripper's pose is tracked using a RealSense T265 camera, and the parallel jaw opening is estimated via ArUco marker tracking. A uniaxial force sensor mounted on the gripper measures grasping force. All sensors operate at 30 Hz. The model is trained to predict 32 steps into the future relative gripper pose, relative jaw position, and absolute grasping force, given inputs of three RGB images from three camera views. During deployment, the robot follows the predicted gripper pose and jaw position for 24 of the 32 predicted steps using impedance control. To prevent the robot from grasping objects with insufficient force, a small gripper closing correction is applied if the predicted gripper force during rollout exceeds the actual measured force by more than 300 grams. The object sets and experimental setups are shown in Figures 7-13.

Examples of alignment between video predictions and real-world executions during the Pick and Place task are illustrated in Figures 14 and 15, highlighting the video policy's ability to generate coherent visual predictions and corresponding robot actions. Additional qualitative results are available in the supplementary video.

Training Background               Unseen Backgrounds

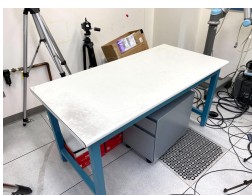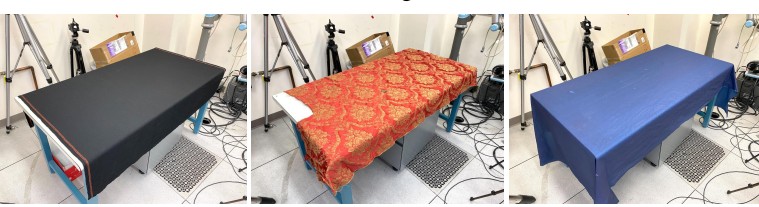

Figure 7: Unseen backgrounds set. In the training set, the four tasks including Pick and Place, M&Ms to Cup, Upright Object and Stack Cups are performed on a white table. In the unseen backgrounds set, the same tasks are evaluated on tables covered with black, red, and blue cloths to test generalization to novel background colors.

Training Set /
Vary Object Location Set            Unseen Objects Set            Unseen Background Set

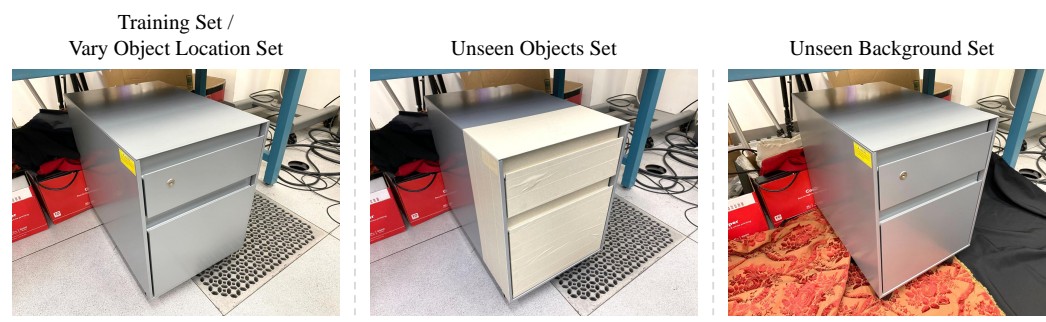

Figure 8: Open Drawer object set. The training and vary object location sets include interactions with both the upper and lower drawers of a single cabinet. In the unseen objects set, the cabinet is covered with masking tape to simulate a different appearance. In the unseen background set, the floor is replaced with a constant red and black fabric.

Training Set /
Vary Object Location Set                                    Unseen Objects Set

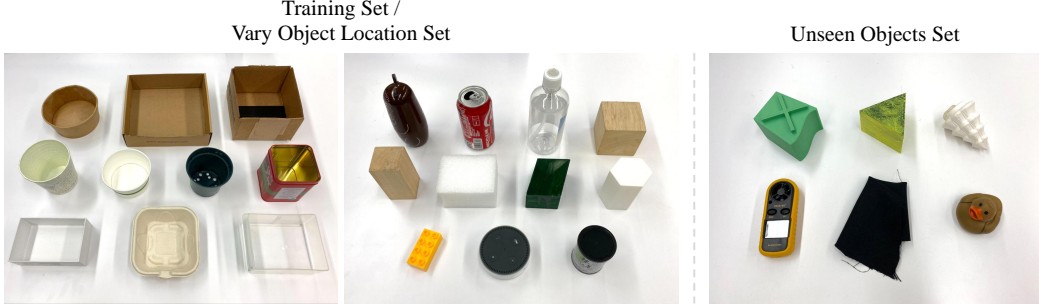

Figure 9: Pick and Place object set. The training set consists of 10 containers and 11 objects. The testing set includes 4 objects with novel shapes and colors to evaluate object generalization.

Training Set /
Vary Object Location Set                                    Unseen Objects Set

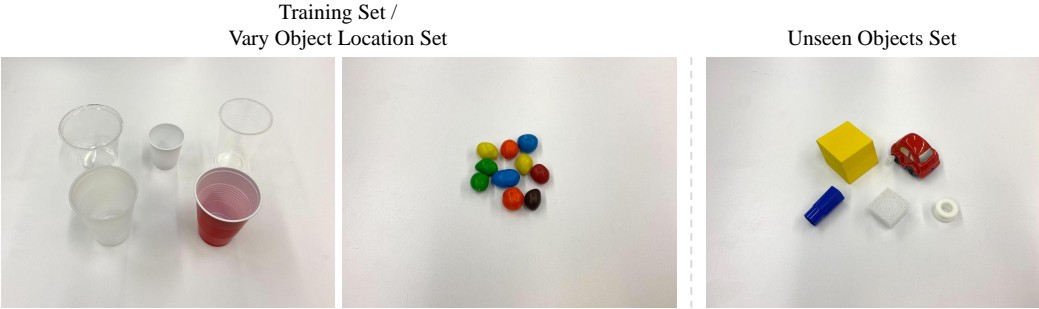

Figure 10: M&Ms to Cup object set. The training set includes 5 cups and M&Ms in 6 different colors. The testing set contains 5 novel objects with varying shapes and colors to assess generalization to unseen items.

Training Set /
Vary Object Location Set

Unseen Objects Set

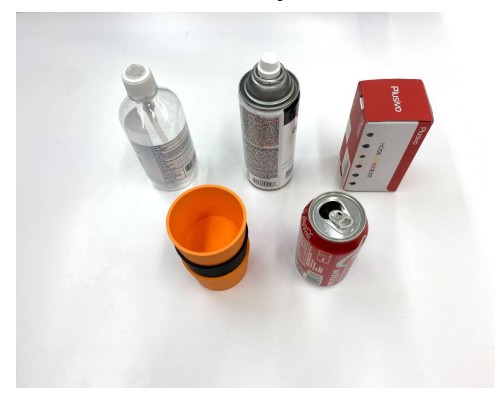

Figure 11: Upright object set. The training set includes 6 objects and the testing set contains 5 objects with novel shapes and colors to evaluate object generalization.

Training Set /
Vary Object Location Set

Unseen Objects Set

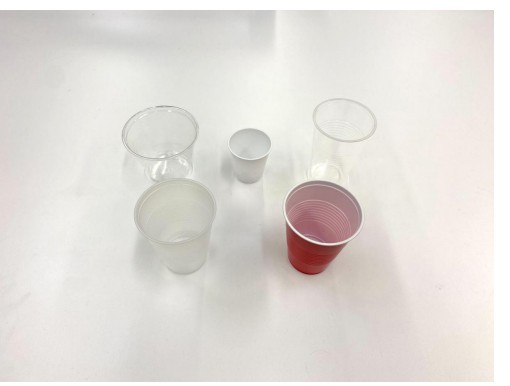
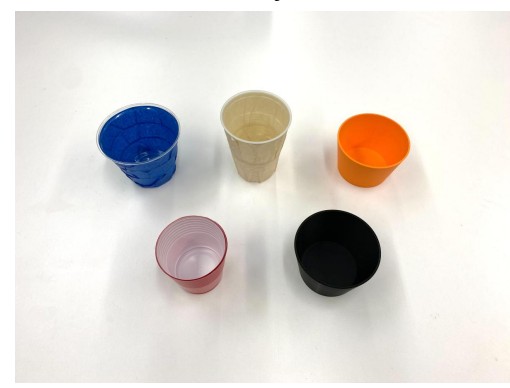

Figure 12: Stack Cups object set. The training set includes 5 cups and the testing set contains 5 cups with novel shapes and colors to evaluate object generalization.

Human Data Collection Setup

Robot Experiment Setup

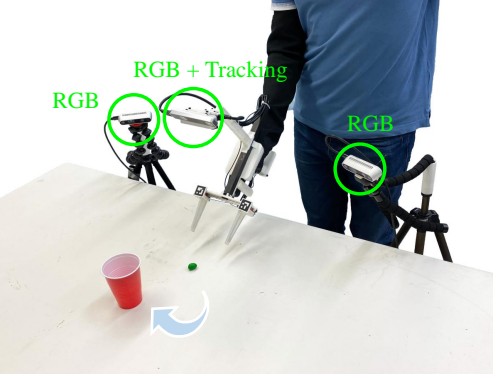
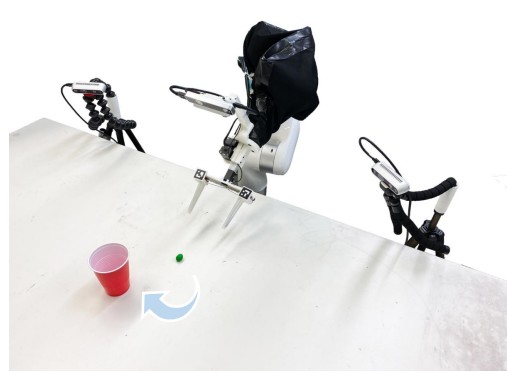

Figure 13: Data collection and robot experiment setup. During data collection, a human demonstrator performs the task using a modified robot gripper. In the robot experiment, the robot executes the tasks with the same setup.

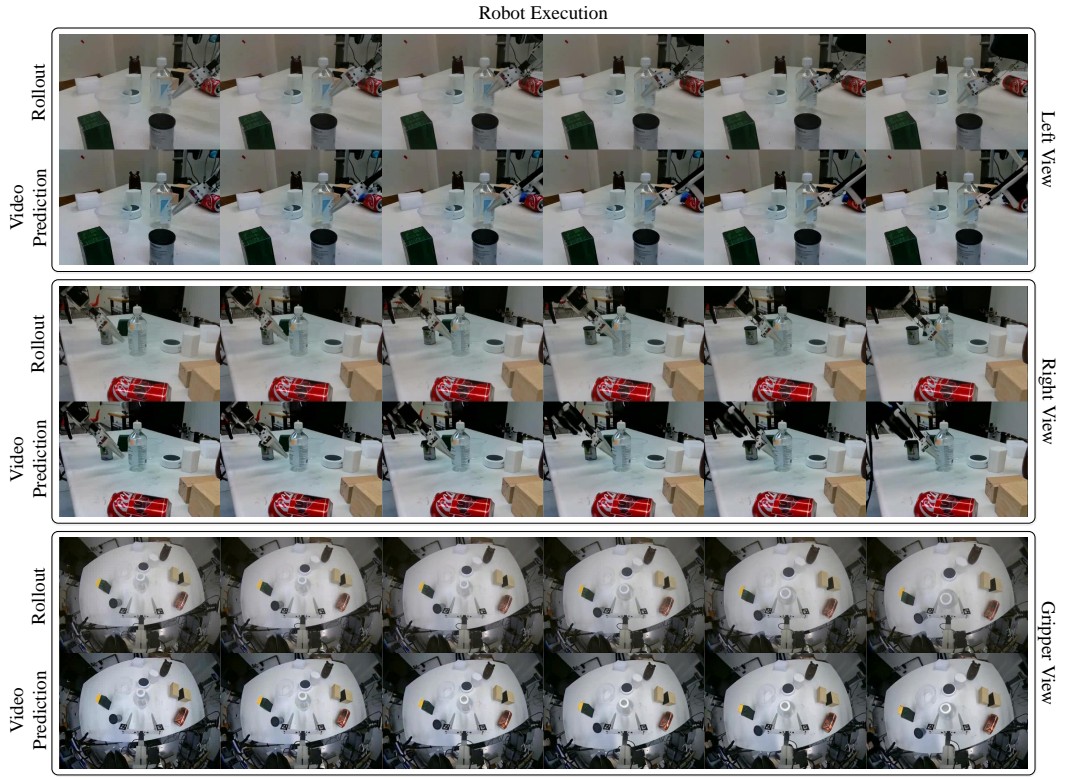

Figure 14: Comparison between video prediction and real-world rollout. For an example Pick and Place task while grasping the object, we show the three camera views during video prediction alongside the corresponding real-world rollout. The alignment between the predicted video and the resulting robot actions demonstrates the effectiveness of Video Policy in generating coherent video and actions.

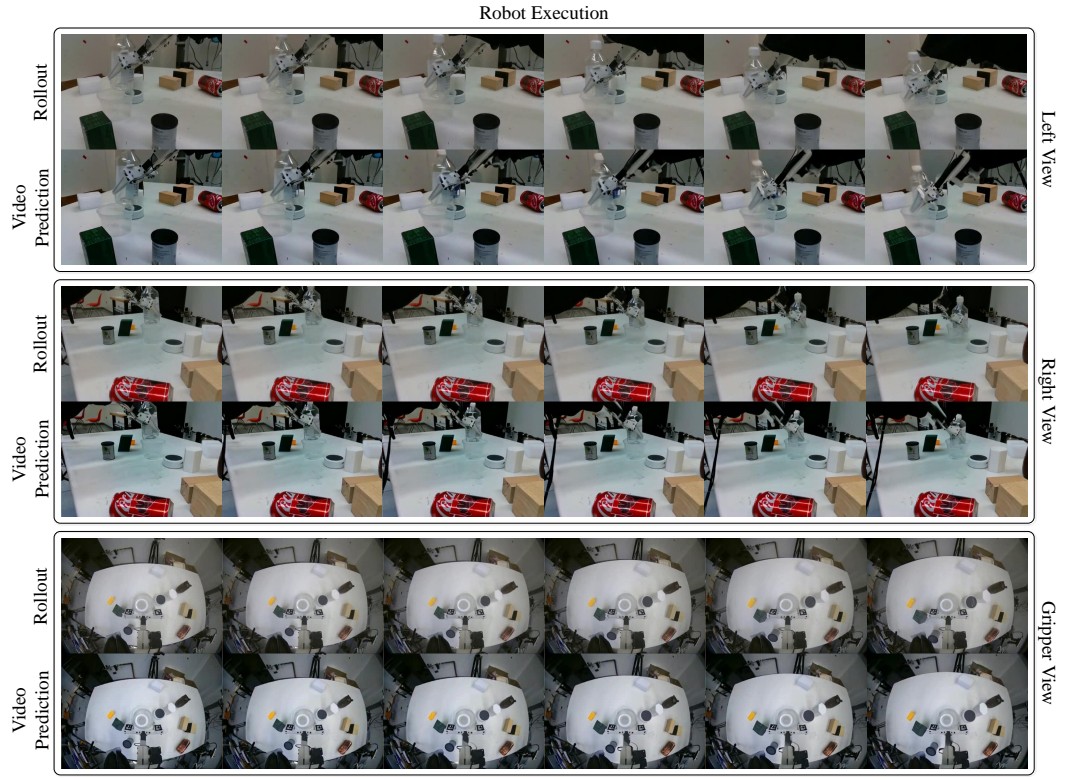

Figure 15: Comparison between video prediction and real-world rollout. For an example Pick and Place task while placing the object into the container, we show the three camera views during video prediction alongside the corresponding real-world rollout. The alignment between the predicted video and the resulting robot actions demonstrates the effectiveness of Video Policy in generating coherent video and actions.