# OpenReview forum: "Video Generators are Robot Policies"
_ICLR.cc/2026/Conference — ICLR 2026 Conference Withdrawn Submission_

### Official Review · Reviewer_6Bm3 · 2025-10-31

**Soundness:** 3
**Presentation:** 3
**Contribution:** 2
**Rating:** 4
**Confidence:** 4

**Summary:**

This paper proposes Video Policy, a framework that treats video generation as a proxy for robot policy learning. Instead of training visuomotor policies purely from demonstrations, the authors show that video diffusion models can implicitly encode generalizable robot behavior, and a lightweight action decoder can map generated video representations into executable robot actions.

**Strengths:**

1. Strong empirical results, especially on RoboCasa and Libero10, showing improved success rates over baselines with a compact architecture.
2. Clarity and reproducibility: the paper is clearly written, with transparent architecture figures, ablations, and hyperparameter details.

**Weaknesses:**

1. While the video diffusion model can be pretrained without actions, the action model still requires action-labeled data for fine-tuning. Therefore, the framework does not eliminate the need for action supervision. The title and framing (“action-free video learning”) are somewhat overstated.
2. The action decoder is trained on limited demonstrations and does not inherit the generalization ability of the video diffusion model. As a result, its performance still depends heavily on the diversity and scale of the action-labeled subset, meaning the method’s scalability is only partial.
3. The design closely resembles previous models such as Video Prediction Policy. Both of which also use video-conditioned architectures followed by action decoders.

**Questions:**

1. The paper describes the video pretraining as action-free, yet the action diffusion model still depends on ground-truth action data. Could the authors clarify to what extent the approach actually reduces the reliance on action supervision compared to prior video-policy methods?
2. Could the authors specify what architectural or training innovations account for their significant performance improvement? Is it primarily due to larger pretrained video priors or improved two-stage optimization?

---

### Official Review · Reviewer_hZje · 2025-10-31

**Soundness:** 3
**Presentation:** 4
**Contribution:** 2
**Rating:** 6
**Confidence:** 4

**Summary:**

This paper proposes Video Policy, a framework that treats video generation as a proxy for robot policy learning. The central insight is that a video generative model, implicitly encodes policy information. A lightweight action decoder can then translate the video model’s latent dynamics into executable robot actions. The architecture jointly trains a video U-Net (adapted from Stable Video Diffusion, SVD) and an action U-Net that predicts robot end-effector actions conditioned on intermediate video features. The paper further highlights that action-free video data can enhance generalization to unseen tasks, suggesting that generative video modeling itself serves as a policy.

**Strengths:**

- The paper is well-written. The introduction and method are easy to follow.

- The experiments are very comprehensive to support the main claims in the paper.

- The video prediction results look similar to the real world.

- The results show that Video Policy is superior to baselines.

- Using action-free data provides a potentially scalable way for data-driven robot learning.

**Weaknesses:**

- The computation cost is high. The inference speed of the video model could slow down the policy rollout. Can the authors provide a comparison between Video Policy and other policy baselines? Can the authors propose some ways to accelerate the policy FPS?
- It remains unclear if Video Policy still performs well in tasks with higher dynamics. The video model may show physics-inplausible results. It’s interesting if the authors can explore the behaviour of the policy and the video prediction model in more dynamic tasks.
- Using more recent art in video models may improve the policy performance.
- The idea of treating video prediction as a robot policy has already been proven working by some earlier work such as (Shuang et al,”Unified video action model.”). Can the authors provide some comparisons with these previous works.

**Questions:**

See the weakness.

---

### Official Review · Reviewer_rAra · 2025-11-01

**Soundness:** 2
**Presentation:** 3
**Contribution:** 2
**Rating:** 2
**Confidence:** 4

**Summary:**

This paper proposes "Video Policy," a framework for learning robot policies by leveraging pre-trained video generation models. The method uses a dual U-Net architecture: a fine-tuned Stable Video Diffusion (SVD) model generates future video frames, and a smaller action decoder, conditioned on the video model's internal features, generates corresponding actions. The central claim is that a two-stage training process—where the video model is frozen before training the action decoder—is superior to end-to-end joint training. The authors present results on the RoboCasa and Libero10 benchmarks, claiming state-of-the-art performance and high sample efficiency with as few as 50 demonstrations.

However, I think the overall novelty of the paper is largely diluted by many related works that guide robot policy via video generation, and the main finding is not original as well.

**Strengths:**

**Strong reported performance on benchmarks**: The paper reports high success rates on the RoboCasa (63% average) and Libero10 (94% average) benchmarks, outperforming several baselines, including some large-scale Vision-Language-Action (VLA) models, while using significantly less demonstration data. These results provide initial evidence for the potential of leveraging powerful video priors.

**Good video generation quality**: The authors give a good demonstration of the capability of the video generation model, which works well in most cases.

**Good representation**: The paper delivery is clear, and the details are available for understanding the proposed method.

**Weaknesses:**

**Overstated Novelty and Lack of Meaningful Comparison to Concurrent Work:** The paper frames the use of video models for policy learning as a novel contribution. This is a significant overstatement. The 2024-2025 period has seen a proliferation of work on this exact topic, including but not limited to concurrent models like **UVA**[1] , **UWM**[2] , and **VPP**[3] , which explore unified video-action architectures. The paper fails to properly situate itself within this crowded landscape, and its core novelty is limited to the specific finding about two-stage training, which is not so surprising, as action finetuning from a  pretrained video backbone seems a natural idea.

**Prohibitive Computational Cost:** The reported inference time of **9 seconds for a single action sequence on an A100 GPU** is a severe limitation that is understated by the authors. This latency makes the system completely unsuitable for real-time, closed-loop control, which is a prerequisite for robotics in any dynamic environment. Dismissing this as a general problem that future hardware will solve is insufficient; it is a fundamental constraint on the proposed method's viability.

**Over dependent on the Video Generation Results** The policy's complete dependence on the video generator is a critical design flaw. The authors admit that real-world failures are caused by "unrealistic video predictions"—where the model imagines a physically impossible future. This creates a brittle, open-loop system. The paper proposes no mechanism to detect or recover from these generative failures, making the approach unreliable for practical applications.

**Reference**:

[1] Li, S., Gao, Y., Sadigh, D., & Song, S. (2025). Unified Video Action Model. *ArXiv, abs/2503.00200*.

[2] Zhu, C., Yu, R., Feng, S., Burchfiel, B., Shah, P., & Gupta, A. (2025). Unified World Models: Coupling Video and Action Diffusion for Pretraining on Large Robotic Datasets. *ArXiv, abs/2504.02792*.

[3] Hu, Y., Guo, Y., Wang, P., Chen, X., Wang, Y., Zhang, J., Sreenath, K., Lu, C., & Chen, J. (2024). Video Prediction Policy: A Generalist Robot Policy with Predictive Visual Representations. *ArXiv, abs/2412.14803*.

**Questions:**

**On Architectural Choices:** Why is a modular, causal `Video -> Action` framework fundamentally better than more integrated approaches like UVA's joint latent space or UWM's unified transformer?

**On System Reliability:** Given that policy failures are directly caused by failures in video generation, the current system appears fundamentally brittle. What concrete mechanisms could be implemented to assess the physical plausibility of generated videos and enable the policy to recover from generative failures?

**On Practical Deployment:** An inference time of 9 seconds is prohibitive for real-world robotics. Beyond general optimism about future diffusion model acceleration, what specific architectural or algorithmic changes to *your proposed method* do you see as the most viable path to achieving real-time control?

---

### Note · Authors · 2025-11-14

I have read and agree with the venue's withdrawal policy on behalf of myself and my co-authors.